# GS-WGAN: A Gradient-Sanitized Approach for Learning Differentially Private Generators

**Dingfan Chen**[1]  **Tribhuvanesh Orekondy**[2]  **Mario Fritz**[1]

[1] CISPA Helmholtz Center for Information Security
[2] Max Planck Institute for Informatics

## Abstract

The wide-spread availability of rich data has fueled the growth of machine learning applications in numerous domains. However, growth in domains with highly-sensitive data (e.g., medical) is largely hindered as the private nature of data prohibits it from being shared. To this end, we propose *Gradient-sanitized Wasserstein Generative Adversarial Networks* (GS-WGAN), which allows releasing a sanitized form of the sensitive data with rigorous privacy guarantees. In contrast to prior work, our approach is able to distort gradient information more precisely, and thereby enabling training deeper models which generate more informative samples. Moreover, our formulation naturally allows for training GANs in both centralized and federated (i.e., decentralized) data scenarios. Through extensive experiments, we find our approach consistently outperforms state-of-the-art approaches across multiple metrics (e.g., sample quality) and datasets.

## 1 Introduction

Releasing statistical and sensory data to a broad community has contributed towards advances in numerous machine learning (ML) techniques e.g., object recognition (ImageNet [11]), language modeling (RCV [24]), recommendation systems (Netflix ratings [6]). However, in many sensitive domains (e.g., medical, financial), similar advances are often held back as the private nature of collected data prohibits release in its original form. Privacy-preserving data publishing [5, 13, 17] provides a reasonable solution, where only a sanitized form of the original data (with rigorous privacy guarantees) is publicly released.

Traditionally, sanitization is performed in a differentially private (DP) framework [12]. The sanitization method employed is often hand-crafted for the given input data [28, 30, 43] and the specific data-dependent task the sanitized data is intended for (e.g., answering linear queries) [7, 14, 21, 36]. As a result, such sanitization techniques greatly restrict the expressiveness of the released data distribution and fail to generalize to novel tasks unanticipated by the publisher. Instead, recent privacy-preserving techniques [5, 41, 42, 44] build on top of successes in generative adversarial network (GANs) [18] literature, to generate synthetic data faithful to the original input distribution. Specifically, GANs are trained using a privacy-preserving algorithm (e.g., using DP-SGD [1]) and demonstrate promising results in modeling a variety of real-world high-dimensional data distributions. Common to most privacy-preserving training algorithms for neural network models is *manipulating* the gradient information generated during backpropagation. Manipulation most commonly involves clipping the gradients (to bound sensitivity) and adding calibrated random noise (to introduce stochasticity). Although recent techniques that employ such an approach demonstrate reasonable success, they are mostly limited to shallow networks and fail to sufficiently capture the sample quality of the original data.

In this paper, towards the goal of a generative model capable of synthesizing high-quality samples in a privacy-preserving manner, we propose a differentially private GAN. We first identify that in

such a data-publishing scenario, only a subset of the trained model (specifically the generator) and its parameters need to be publicly-released. This insight allows us to surgically manipulate the gradient information during training, and thereby allowing more meaningful gradient updates. By coupling the approach with a Wasserstein [2] objective with gradient-penalty term [19], we further improve the amount of gradient information flow during training. The Wasserstein objective additionally allows us to precisely estimate the gradient norms and analytically determine the sensitivity values. As an added benefit, we find our approach bypasses an intensive and fragile hyper-parameter search for DP-specific hyperparameters (particularly clipping values).

**Contributions.** *(i)* A novel gradient-sanitized Wasserstein GAN (GS-WGAN), which is capable of generating high-dimensional data with DP guarantee; *(ii)* Our approach naturally extends to both centralized and decentralized datasets. In the case of decentralized scenarios, our work can provide user-level DP guarantee [26] under an untrusted server; *(iii)* Extensive evaluations on various datasets demonstrate that our method significantly improves the sample quality of privacy-preserving data over state-of-the-art approaches.

## 2 Related Work

We review several differentially private GAN models, as well as their relations to our work.

**DP-SGD GAN.** Training GANs via DP-SGD [1, 5, 16, 38, 41, 44] has proven effective in generating high-dimensional sanitized data. However, DP-SGD relies on carefully tuning of the clipping bound of gradient norm, i.e., the sensitivity value. Specifically, the optimal clipping bound varies greatly with the model architecture and the training dynamics, making the implementation of DP-SGD difficult. Unlike previous works, we selectively apply sanitization to a necessary and sufficient subset of gradients for preserving privacy, which enables us to exploit the theoretical property of Wasserstein GANs [2, 19] for a precise estimation of the sensitivity value, avoiding the intensive search of hyper-parameters while reducing the clipping bias.

**PATE.** Private Aggregation of Teacher Ensembles (PATE) is recently adapted to generative models and two main approaches were studied: PATE-GAN [42] and G-PATE [29]. PATE-GAN trained multiple teacher discriminators on disjoint data partitions together with a student discriminator. In contrast, we consider a simplified model without a student discriminator.

G-PATE [29] is similar to our work in the sense that, both works trained the discriminator non-privately while only training the generator with DP guarantee, and both sanitized gradients that the generator received from the discriminator. However, G-PATE suffers from two main limitations: *(i)* gradients need to be discretized by using manually selected bins in order to suit for the PATE framework and *(ii)* high-dimensional gradients in the PATE framework bring high privacy costs and thus dimension reduction techniques are required. Our framework can effectively avoid these two limitations and achieve better sample quality due to the novel gradient sanitation, see our experiments.

**Fed-Avg GAN [3].** While many works focus on centralized setting, the decentralized case has rarely been studied. To address this, Federated Average GAN (Fed-Avg GAN) proposed to adapt GAN training by using the DP-Fed-Avg [31] algorithm, providing user-level DP guarantee under trusted server. In comparison with Fed-Avg GAN that merely works on decentralized data, our work can tackle both centralized and decentralized data using a single framework. Note that Fed-Avg sanitized parameter gradients of the discriminator in a similar way to DP-SGD, it also suffers from the difficulty of turning hyper-parameters.

## 3 Background

DP provides rigorous privacy guarantees for algorithms while allowing for quantitative privacy analysis. We below present several definitions and theorems that will be used in this work.

**Definition 3.1.** (Differential Privacy (DP) [12]) A randomized mechanism $\mathcal{M}$ with range $\mathcal{R}$ is $(\varepsilon, \delta)$-DP, if

$$Pr[\mathcal{M}(S) \in \mathcal{O}] \leq e^{\varepsilon} \cdot Pr[\mathcal{M}(S') \in \mathcal{O}] + \delta \tag{1}$$

holds for any subset of outputs $\mathcal{O} \subseteq \mathcal{R}$ and for any adjacent datasets $S$ and $S'$, where $S$ and $S'$ differ from each other with only one training example. $\mathcal{M}$ is the GAN training algorithm in our case, $\varepsilon$

corresponds to the upper bound of privacy loss, and $\delta$ is the probability of breaching DP constraints. Intuitively, DP guarantees the difficulty of inferring the presence of an individual in the private dataset by observing $\mathcal{M}(S)$.

**Definition 3.2.** (Rényi Differential Privacy (RDP) [33]) A randomized mechanism $\mathcal{M}$ is $(\lambda, \varepsilon)$-RDP with order $\lambda$, if

$$D_\lambda(\mathcal{M}(S)\|\mathcal{M}(S')) = \frac{1}{\lambda-1}\log \mathbb{E}_{x\sim\mathcal{M}(S)}\left[\left(\frac{Pr[\mathcal{M}(S)=x]}{Pr[\mathcal{M}(S')=x]}\right)^{\lambda-1}\right] \leq \varepsilon \tag{2}$$

holds for any adjacent datasets $S$ and $S'$, where $D_\lambda(P\|Q) = \frac{1}{\lambda-1}\log \mathbb{E}_{x\sim Q}[(P(x)/Q(x))^\lambda]$ denotes the Rényi divergence. Moreover, a $(\lambda, \varepsilon)$-RDP mechanism $\mathcal{M}$ is also $(\varepsilon + \frac{\log 1/\delta}{\lambda-1}, \delta)$-DP.

In contrast to DP, RDP provides convenient composition properties to accumulate privacy cost over a sequence of mechanisms (i.e., multiple gradient descent steps in our case).

**Theorem 3.1.** (Composition) For a sequence of mechanisms $\mathcal{M}_1, ..., \mathcal{M}_k$ s.t. $\mathcal{M}_i$ is $(\lambda, \varepsilon_i)$-RDP $\forall i$, the composition $\mathcal{M}_1 \circ ... \circ \mathcal{M}_k$ is $(\lambda, \sum_i \varepsilon_i)$-RDP.

Our approach is built on top of the Gaussian mechanism defined as follows.

**Definition 3.3.** (Gaussian Mechanism [15, 33]) Let $f : X \to \mathbb{R}^d$ be an arbitrary $d$-dimensional function with sensitivity being

$$\Delta_2 f = \max_{S,S'} \|f(S) - f(S')\|_2 \tag{3}$$

over all adjacent datasets $S$ and $S'$. The Gaussian Mechanism $\mathcal{M}_\sigma$, parameterized by $\sigma$, adds noise into the output,i.e.,

$$\mathcal{M}_\sigma(x) = f(x) + \mathcal{N}(0, \sigma^2 I). \tag{4}$$

$\mathcal{M}$ is $(\lambda, \frac{\lambda\Delta_2 f^2}{2\sigma^2})$-RDP.

To provide DP guarantees of the released generator, we exploit the closedness of DP under post-processing, which is formalized as the following theorem.

**Theorem 3.2.** (Post-processing [15]) If $\mathcal{M}$ satisfies $(\varepsilon, \delta)$-DP, $F \circ \mathcal{M}$ will satisfy $(\varepsilon, \delta)$-DP for any function $F$ with $\circ$ denoting the composition operator.

# 4 Proposed Method

**Generative Adversarial Networks (GANs) [18].** Our approach models the underlying (private) data distribution using a generative neural network, building on top of recent successes of GANs. GANs (see Fig. 1(a)) formulate the task of sample generation as a zero-sum two-player game, between two neural network models: discriminator $D$ and generator $G$. The discriminator $D$ is rewarded for correctly classifying whether a given sample is 'real' (i.e., from the input data distribution) or 'fake' (generated by the generator). In contrast, the task of the generator $G$ is (given some random noise $z$) to generate samples which fool the discriminator (i.e., causes misclassifications). After training the models in an adversarial manner, the discriminator is discarded and the generator is used as a proxy to draw samples from the original distribution.

**Differentially Private GANs.** Releasing the generator as a substitute for the original training data distribution entails privacy risks [10]. Consequently, along the lines of recent work [5, 38, 41, 44], our goal is instead to train the GAN in a privacy-preserving manner, such that any privacy leakage upon disclosing the generator is bounded. A simple approach towards the goal is replacing the typical training procedure (SGD) with a differentially private variant (DP-SGD [1]) and thereby limiting the contribution of a particular training example in the final trained model. DP-SGD enforces the desired privacy requirement by *(i)* clipping the gradients $g_t$ to have an $L_2$-norm no larger than $C$ at each training step; and *(ii)* sampling random noise and adding it to the gradients, before performing descent on the trained parameters $\theta$:

$$g^{(t)} := \nabla_\theta \mathcal{L}(\theta_D, \theta_G) \qquad \text{(gradient)} \tag{5}$$

$$\hat{g}^{(t)} := \mathcal{M}_{\sigma,C}(g^{(t)}) = \text{clip}(g^{(t)}, C) + \mathcal{N}(0, \sigma^2 C^2 I) \qquad \text{(sanitization mechanism)} \tag{6}$$

$$\theta^{(t+1)} := \theta^{(t)} - \eta \cdot \hat{g}^{(t)} \qquad \text{(gradient descent step)} \tag{7}$$

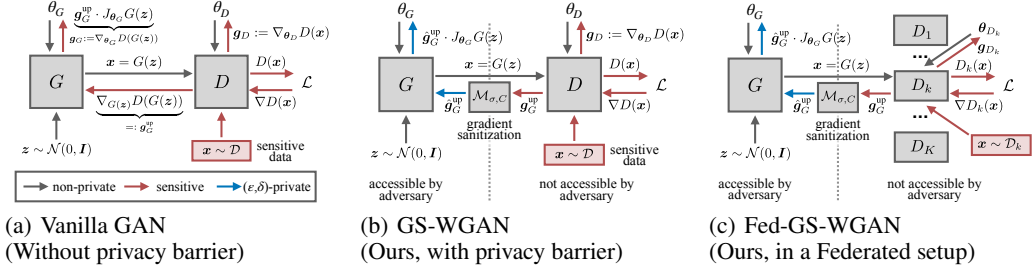

(a) Vanilla GAN
(Without privacy barrier)

(b) GS-WGAN
(Ours, with privacy barrier)

(c) Fed-GS-WGAN
(Ours, in a Federated setup)

**Figure 1:** Approach outline. Our gradient sanitization scheme ensures DP training of the generator.

While such an approach provides rigorous privacy guarantees, there are multiple shortcomings: *(i)* the sanitization mechanism $\mathcal{M}_{\sigma,C}$, primarily due to clipping, significantly destroys the original gradient information, and thereby affects utility; and *(ii)* finding a reasonable clipping value $C$ in the mechanism to balance utility with privacy is especially challenging. In particular, as the gradient norms exhibit a heavy-tailed distribution, choosing a clipping value requires an exhaustive search. Moreover, since the clipping value is extremely sensitive to many other hyperparameters (e.g., learning rate, architecture), it requires persistent re-tuning. Now, we discuss how we address these shortcomings within our gradient-sanitized approach.

**Selectively applying Sanitization Mechanism.** We begin by exploiting the fact that after training the GAN, only the generator $G$ is released. Consequently, we can perform gradient steps by selectively applying the sanitization mechanism only to the corresponding subset of parameters $\boldsymbol{\theta}_G$:

$$\boldsymbol{\theta}_D^{(t+1)} := \boldsymbol{\theta}_D^{(t)} - \eta_D \cdot \boldsymbol{g}_D^{(t)} \qquad (\hat{\boldsymbol{g}}_D^{(t)} = \boldsymbol{g}_D^{(t)}; \text{Discriminator}) \qquad (8)$$

$$\boldsymbol{\theta}_G^{(t+1)} := \boldsymbol{\theta}_G^{(t)} - \eta_G \cdot \hat{\boldsymbol{g}}_G^{(t)} \qquad (\hat{\boldsymbol{g}}_G^{(t)} = \mathcal{M}_{\sigma,C}(\boldsymbol{g}_G^{(t)}); \text{Generator}) \qquad (9)$$

Apart from reducing the number of parameters sanitized, this also provides a benefit of more reliably training a discriminator. In addition, we exploit the chain rule to further narrow the scope of the sanitization mechanism:

$$\boldsymbol{g}_G = \nabla_{\boldsymbol{\theta}_G} \mathcal{L}_G(\boldsymbol{\theta}_G) = \nabla_{G(\boldsymbol{z};\boldsymbol{\theta}_G)} \mathcal{L}_G(\boldsymbol{\theta}_G) \cdot J_{\boldsymbol{\theta}_G} G(\boldsymbol{z};\boldsymbol{\theta}_G) \qquad (10)$$

$$\hat{\boldsymbol{g}}_G = \mathcal{M}_{\sigma,C}(\underbrace{\nabla_{G(\boldsymbol{z})} \mathcal{L}_G(\boldsymbol{\theta}_G)}_{\boldsymbol{g}_G^{\text{upstream}}}) \cdot \underbrace{J_{\boldsymbol{\theta}_G} G(\boldsymbol{z};\boldsymbol{\theta}_G)}_{\boldsymbol{J}_G^{\text{local}}} \qquad (11)$$

The above becomes easier to intuit by considering a typical loss function $\mathcal{L}_G(\boldsymbol{\theta}_G) = -D(G(\boldsymbol{z};\boldsymbol{\theta}_G))$. As illustrated in Fig. 1(b), Eq. 11 can then be considered as placing the privacy barrier for gradient information backpropagating from the discriminator back to the generator, by applying the sanitization mechanism on $\boldsymbol{g}_G^{\text{upstream}}$. Note that the second term ($\boldsymbol{J}_G^{\text{local}}$) is the local generator jacobian computed independent of training data, and hence does not require sanitization. Consequently, using a more precise application of the sanitization mechanism on the gradient information, our goal here is to maximally preserve the true gradient direction during training.

**Bounding sensitivity using Wasserstein distance.** To bound the sensitivity of the optimizer on individual training examples, a key step in sanitization mechanisms is to *clip* (Eq. 6) the gradient vector $\boldsymbol{g}$ (Eq. 5) before updating parameters (Eq. 7). Clipping is typically performed in $L_2$ norm, by replacing the gradient vector $\boldsymbol{g}$ by $\boldsymbol{g}/\max(1, \|\boldsymbol{g}\|_2/C)$ to ensure $\|\boldsymbol{g}\|_2 \leq C$. However, clipping significantly destroys gradient information, as reasonable choices of $C$ (e.g., 4 [1]) are significantly lower than the gradient-norms observed ($12 \pm 10$ in our case) when training neural networks using standard loss functions. We propose to alleviate the issue by leveraging a more suitable loss function, which generates bounded gradients (with norms close to 1) by construction. Specifically, we use as our loss the Wasserstein-1 metric [2], which measures the statistical distance between the real and generated data distributions. Here, the training process can be interpreted as minimizing integral probability metrics (IPMs) $\sup_{f \in \mathcal{F}} |\int_M f dP - \int_M f dQ|$ between real ($P$) and generated ($Q$) data distributions, where $\mathcal{F} = \{f : \|f\|_L \leq 1\}$ (i.e., the discriminator function $f$ is 1-Lipschitz continuous). Theoretically, the optimal discriminator has a gradient norm being 1 almost everywhere under $P$ and $Q$ [19] (i.e., $\|\boldsymbol{g}_G^{\text{upstream}}\|_2 \approx 1$).

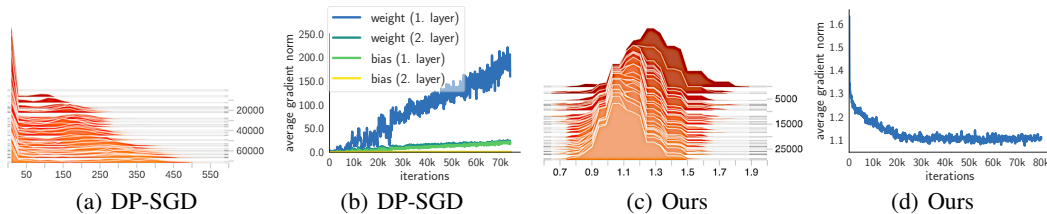

| (a) DP-SGD | (b) DP-SGD | (c) Ours | (d) Ours |

**Figure 2:** Gradient norm (before clipping) dynamics during the GAN training process. In the experiment, the clipping bound is chosen to be 1 and 1.1 in 2(c) and 2(a) respectively.

We incorporate the norm constraint into our training objective in the form of a gradient penalty term [19]:

$$\mathcal{L}_D = -\mathbb{E}_{\boldsymbol{x} \sim P}[D(\boldsymbol{x})] + \mathbb{E}_{\tilde{\boldsymbol{x}} \sim Q}[D(\tilde{\boldsymbol{x}})] + \lambda \mathbb{E}[(\|\nabla D(\alpha \boldsymbol{x} + (1-\alpha)\tilde{\boldsymbol{x}})\|_2 - 1)^2] \qquad (12)$$
$$\mathcal{L}_G = -\mathbb{E}_{\boldsymbol{z} \sim P_{\boldsymbol{z}}}[D(G(\boldsymbol{z}))] \qquad (13)$$

where $\mathcal{L}_D$ and $\mathcal{L}_G$ represent training objectives for the discriminator and the generator, respectively. $\lambda$ is the hyper-parameter for weighting the gradient penalty term and $P_{\boldsymbol{z}}$ denotes the prior distribution for the latent code variable $\boldsymbol{z}$. The variable $\alpha \sim \mathcal{U}[0,1]$, uniformly sampled from $[0,1]$, regulates the interpolation between real and generated samples.

As a natural consequence of the Wasserstein objective, bounding the norms of our target gradient $\boldsymbol{g}_G^{\text{upstream}}$ ( Equation 11) during training is integrated in our training objective (last term in Equation 12). Consequently, we observe significantly lower variance in gradient norms during training (see Fig. 2(c)-2(d)) compared to training using a standard GAN loss (see Fig. 2(a)-2(b)). As a result, bounding the sensitivity (gradient norms) is now largely delegated to our training procedure and clipping using the sanitization mechanism destroys significantly less information. Additionally, we obtain the optimal clipping threshold of $C$=1, as $\|\boldsymbol{g}_G^{\text{upstream}}\|_2 \approx 1$ based on the theoretical property of Wasserstein GANs. This allows us to derive a fixed and bounded sensitivity, eliminating the need for intensive hyper-parameter search for a proper clipping threshold. Following this clipping strategy, a data-independent privacy cost can be determined by the following theorem, whose proof is provided in Appendix.

**Theorem 4.1.** Each generator update step satisfies $(\lambda, 2B\lambda/\sigma^2)$-RDP where $B$ is the batch size.

**Privacy Amplification by Subsampling.** A well-known approach for increasing privacy of a mechanism is to apply the mechanism to a random subsample of the database, rather than on the entire dataset [4, 27, 39]. Intuitively, subsampling decreases the chances of leaking information about a particular individual since nothing about that individual can be leaked once the individual is not included in the subsample. In order to further reduce the privacy cost, we subsample the whole dataset into different subsets and train multiple discriminators independently on each subset. At each training step, the generator randomly queries one discriminator while the selected discriminator updates its parameters on the generated data and its associated subsampled dataset.

**Extending to Federated Learning.** In addition to improving the privacy guarantee, performing subsampling in our setup also naturally accommodates training a generative model on decentralized datasets (with a discriminator trained on each disjoint data subset). Recently, Augenstein et al. [3] identified such techniques are extremely relevant when training models in a federated setup [32], i.e., when the training data is private and distributed among edge devices. We outline our method to train a differentially private GAN in a federated setup in Figure 1(c) and remark some subtle differences between our approach and Fed-Avg GAN [3] here: *(i)* the discriminators are retained at each client in our framework while they are shared between the server and client in Fed-Avg GAN; *(ii)* the gradients are sanitized at each client before sending to the server, with which we provide DP guarantee even under an untrusted server. In contrast, the unprocessed information is accumulated at the server before being sanitized in Fed-Avg GAN; and *(iii)* The gradients w.r.t. the samples are transferred in GS-WGAN, while Fed-Avg GAN transfers the gradients w.r.t. discriminator network parameters.

|  |  | IS↑ | FID↓ | MLP↑ Acc | CNN↑ Acc | Avg↑ Acc | Calibrated↑ Acc |
|---|---|---|---|---|---|---|---|
| MNIST | Real | 9.80 | 1.02 | 0.98 | 0.99 | 0.88 | 100 % |
|  | G-PATE [1] | 3.85 | 177.16 | 0.25 | 0.51 | 0.34 | 40% |
|  | DP-SGD GAN | 4.76 | 179.16 | 0.60 | 0.63 | 0.52 | 59% |
|  | DP-Merf | 2.91 | 247.53 | 0.63 | 0.63 | 0.57 | 66% |
|  | DP-Merf AE | 3.06 | 161.11 | 0.54 | 0.68 | 0.42 | 47% |
|  | Ours | **9.23** | **61.34** | **0.79** | **0.80** | **0.60** | **69%** |
| Fashion-MNIST | Real | 8.98 | 1.49 | 0.88 | 0.91 | 0.79 | 100% |
|  | G-PATE | 3.35 | 205.78 | 0.30 | 0.50 | 0.40 | 54% |
|  | DP-SGD GAN | 3.55 | 243.80 | 0.50 | 0.46 | 0.43 | 53% |
|  | DP-Merf | 2.32 | 267.78 | 0.56 | 0.62 | 0.51 | 65% |
|  | DP-Merf AE | 3.68 | 213.59 | 0.56 | 0.62 | 0.45 | 55% |
|  | Ours | **5.32** | **131.34** | **0.65** | **0.65** | **0.53** | **67%** |

**Table 1:** Quantitative Results on MNIST and Fashion-MNIST ($\varepsilon = 10, \delta = 10^{-5}$)

## 5 Experiment

### 5.1 Experiment Setup

To validate the applicability of our method to high-dimensional data, we conduct experiments on image datasets. In line with previous works, we use MNIST [23] and Fashion-MNIST [40] dataset. We model the joint distribution of images and the corresponding labels, i.e., the label is supplied to both the generator and the discriminator, and the image is generated conditioned on the input. During both training and inference, we use a uniform prior distribution for generating labels, which is independent of the training dataset and thus does not incur additional privacy cost (in contrast, [38] needs to assume the labels are non-private).

**Evaluation Metrics.** We evaluate along two fronts: *privacy* (determined by $\varepsilon$) and *utility*. For utility, we consider two metrics: (a) *sample quality*: realism of the samples produced – evaluated by **Inception Score (IS)** [25, 37] and **Frechet Inception Distance (FID)** [22] (standard in GAN literature); and (b) *usefulness for downstream tasks*: we train downstream classifiers on 60k privately-generated data points and evaluate the prediction accuracy on real test set. We consider Multi-layer Perceptrons (MLP), Convolutional Neural Networks (CNN) and 11 scikit-learn [34] classifiers (e.g., SVMs, Random Forest). We include the following metrics in the main paper: **MLP Acc** (MLP accuracy), **CNN Acc** (CNN accuracy), **Avg Acc** (Averaged accuracy of all classification models), **Calibrated Acc** (Averaged accuracy of all classification models normalized by the accuracy when trained on real data). The detailed results are presented in Appendix.

**Architecture and Warm-start.** We highlight two strategies adopted for improving the sample quality as well as reducing the privacy cost: *(i) Better model architecture*: While previous works are limited to shallow networks and thereby bottle-necking generated sample quality, our framework allows stable training with a complex model architecture (DCGAN [35] architecture for the discriminator, ResNet architecture (adapted from BigGAN [8]) for the generator) to help improve the sample quality; and *(ii) Discriminator warm-starting*: To bootstrap the training process, we pre-train discriminators along with a non-private generator for a few steps, and we subsequently train the private generator using the warm-starting values of the discriminators. Note that our framework allows pre-training on the original private dataset without compromising privacy (in contrast, [44] needs to use external public datasets).

### 5.2 Comparison with Baselines

**Baselines.** We consider the following state-of-the-art methods designed for DP high-dimensional data generation: **DP-Merf** and **DP-Merf AE** [20], **DP-SGD GAN** [38, 41, 44], and **G-PATE** [29]. While PATE-GAN [42] demonstrates promising results on low-dimensional data, we currently do not consider it as we were unable to extend it to our image datasets (more details in appendix) for

| Method | MNIST | Fashion-MNIST |
|--------|-------|---------------|
| G-PATE | | |
| DP-SGD GAN | | |
| DP-Merf | | |
| DP-Merf AE | | |
| Ours | | |

**Figure 3:** Generated samples with $(\varepsilon, \delta) = (10, 10^{-5})$

a fair comparison. For DP-Merf, DP-Merf AE, and G-PATE, we use the source code provided by the authors. For DP-SGD GAN, we adopt the implementation of [38], which is the only work that provides executable code with privacy analysis. For a fair comparison, we evaluate all methods with a privacy budget of $(\varepsilon, \delta)=(10, 10^{-5})$ (consistently used in previous works) over 60K generated samples.

**Results.** We present the qualitative results in Figure 3 and the quantitative results in Table 1. In terms of sample quality, we find (Table 1, columns IS and FID) our method consistently provides significant improvements over baselines. For instance, considering inception scores, we find a relative improvement of 94% (9.23 vs. 4.76 of DP-SGD GAN) on MNIST and 45% on Fashion-MNIST (5.32 vs. 3.68 of DP-Merf AE).

Furthermore, our method also generates samples that better capture the statistical properties of the original data and are thereby making aiding performances of downstream tasks. For instance, our approach increases performance of a downstream MLP classifier(Table 1, column MLP Acc) by 25% (0.79 vs. 0.63 of DP-Merf) on MNIST and 16% (0.65 vs. 0.56 of DP-Merf) on Fashion-MNIST. In a word, our approach demonstrates significant improvements across multiple metrics and high-dimensional image datasets.

## 5.3 Influence of Hyperparameters

The privacy/utility performances of our approach is primarily determined by three factors:*(i)* **subsampling rates** $\gamma$, *(ii)* number of training **iterations**, and *(iii)* **noise scale** $\sigma$. We now investigate how these factors influence privacy cost $\varepsilon$ and utility (sample quality measured by IS and FID), and additionally compare with baselines:

*(i)* **Subsampling rates**: We evaluate the sample quality of our method considering multiple choices of subsampling rates ($\gamma \in [1/250, 1/500, 1/1000, 1/1500]$) over the training iterations. The results are presented in Figure 4(a), where the $x$-axis corresponds to the $\varepsilon$ value evaluated at different iterations. We observe that the sub-sampling rate should be sufficiently small for achieving a reasonable sample quality while providing a strong privacy guarantee. A value of $1/1000$ yields relatively good privacy-utility trade-off, while further decreasing the sub-sampling rate does not necessarily improve the results. *(ii)* **Iterations**: We evaluate all methods during the course of training, where more iterations lead to higher utilities, but at the expense of accumulating a higher privacy cost $\varepsilon$. From Figure 4(b), we find our approach yields better sample qualities with fewer iterations (and hence lower $\varepsilon$). Specifically, across the range of iterations, we find IS increases by 10-90%, while the

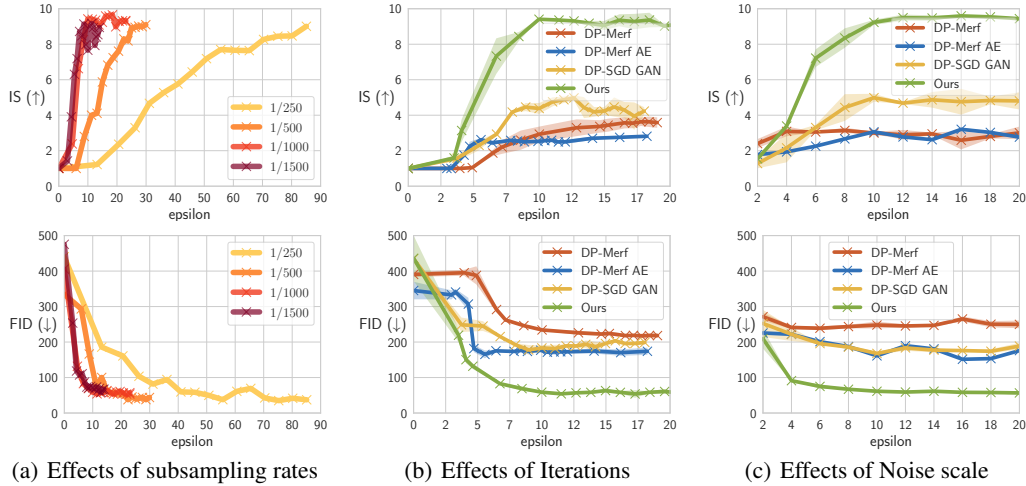

(a) Effects of subsampling rates     (b) Effects of Iterations     (c) Effects of Noise scale

**Figure 4:** Privacy-utility trade-off on MNIST with $\delta = 10^{-5}$. (Top row: IS. Bottom row: FID.)

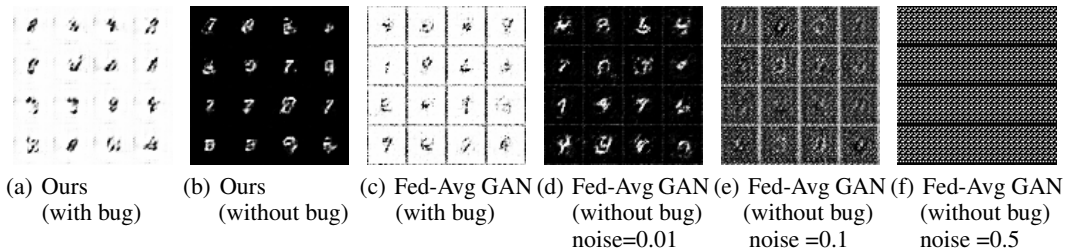

(a) Ours   (b) Ours   (c) Fed-Avg GAN (d) Fed-Avg GAN (e) Fed-Avg GAN (f) Fed-Avg GAN
(with bug)   (without bug)   (with bug)   (without bug)   (without bug)   (without bug)
                                                             noise=0.01       noise =0.1       noise =0.5

**Figure 5:** Qualitative Results on Federated EMNIST.

FID decreases by 20-60% compared to baselines. *(iii)* **Noise scale**: We calibrate the noise scale of each method to certain privacy budget $\varepsilon$ and show the resulting privacy-utility curves in Figure 4(c). Similar to the previous case, our method achieves a consistent improvement in both metrics spanning a broad range of noise scale (privacy budget $\varepsilon$).

## 5.4 Federated Setting Evaluation

Our approach allows to perform privacy-preserving training of a GAN in federated setup, where sensitive user dataset is partitioned across $K$ clients (e.g., edge devices). Such a training scheme is useful to privately inspect

|  | IS ↑ | FID ↓ | epsilon ↓ | CT (byte) ↓ |
|---|---|---|---|---|
| Fed Avg GAN | 10.88 | 218.24 | $9.99 \times 10^6$ | $\sim 3.94 \times 10^7$ |
| Ours | **11.25** | **60.76** | $\mathbf{5.99 \times 10^2}$ | $\sim \mathbf{1.50 \times 10^5}$ |

**Table 2:** Quantitative Results on Federated EMNIST ($\delta = 1.15 \times 10^{-3}$)

data for debugging. For evaluation, we consider a real-world debugging task introduced in [3]: to detect the erroneous flipping of pixel intensities, which occurs in a fraction of client devices. Two GAN models are trained: one on client data that are suspected to be erroneous flipped (with bug) and one on the client data that are believed to be normal (without bug). The samples generated by these two GAN models should exhibit different appearance such that the bug can be detected by inspecting the generated samples. To mimic the real-world situation where the server is blind to the erroneous pre-processing, only a fraction of the suspected users is indeed affected by the bug. This has the realistic property that the client data is non-IID and poses additional difficulties in the GAN training. A detailed description about the data can be found in Appendix.

We conduct experiments on the Federated EMNIST dataset [9] and compare our GS-WGAN with **Fed-Avg GAN** [3].As shown in Figure 5(a) and 5(b), the presence of bug is clearly identifiable by inspecting the samples generated by our model. Moreover, as shown in Table 2, our GS-WGAN yields better sample quality ($0.28\times$ smaller FID) with a significantly lower privacy cost ($10^4\times$ smaller $\varepsilon$) compared to Fed-Avg GAN. Furthermore, our method shows better robustness against large injected noise. This is illustrated in Figure 5(e) and 5(f): a noise scale larger than 0.1 inevitably leads to

failure in training Fed-Avg GAN, whereas our method can tolerate 10 times larger noise scale. In addition, we show in the last column of Table 2 the amortized communication cost (CT) required for performing one update step on the generator. Specifically, this corresponds to the total number of transferred bytes (including both server-to-client and client-to-server) averaged over all participating clients. Our GS-WGAN allows each client to retain its discriminator locally and only the gradients w.r.t. generated samples are communicated (which is significantly more compact than gradients w.r.t model parameters, as done by Fed-Avg GAN). We observe that GS-WGAN achieves a magnitude of $10^2$ gain in reducing the communication cost.

## 6 Conclusion

In this paper, we presente a differentially-private approach *GS-WGAN* to sanitize sensitive high-dimensional datasets with provable privacy guarantees while simultaneously preserving informativeness of the sanitized samples. Our primary insight is that privacy-preserving training (which sacrifices utility) can be selectively applied only to the generator (which is publicly released) while the discriminator (which is discarded post-training) can be trained optimally. Additionally, introducing a Wasserstein training objective allows us to exploit the Lipschitz property of the discriminator and leads to precise estimates of the sensitivity value without exhaustive hyper-parameters search. Our extensive evaluation presents encouraging results: sensitive datasets can be effectively distilled to sanitized forms which nonetheless preserves informativeness of the data and allows training downstream models.

## 7 Broader Impact

The success of many machine learning methods hinges upon the availability of (large) datasets, which is problematic if the data is sensitive and contains private information, e.g., in the health domain, where diagnosis, treatment and personalized medicine are subject to strict privacy constraints. In contrast to direct privacy-preserving analysis, privacy-preserving generative models provide a safe way to release data, yielding several important implications: (1) allowing for wide applications without changing analysis algorithms as a result of sanitized data; (2) promoting new scientific discovery that could be handicapped due to data protection hurdles; (3) providing public benchmarks/datasets in domains with sensitive data to foster fair comparison and reproducible research.

This work contributes to making the latest advances in generative modeling complying with data privacy—a commonly agreed societal value. Our method improves the state of the art in privacy-preserving data generation. In particular, the success of our approach on high-dimensional data shows its potential in a broader range of applications.

## Acknowledgments and Disclosure of Funding

This work is partially funded by the Helmholtz Association within the projects "Trustworthy Federated Data Analytics (TFDA)" (ZT-I-OO1 4) and "Protecting Genetic Data with Synthetic Cohorts from Deep Generative Models (PRO-GENE-GEN)" (ZT-I-PF-5-23).

## Footnotes

[1]  PATE provides data-dependent $\varepsilon$, i.e., publishing $\varepsilon$ value will introduce privacy cost. Thus, G-PATE is not directly comparable to other methods and is excluded from our analysis study (section 5.3).

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
