[Supplementary Material]

# Supplementary Material for "GS-WGAN: A Gradient-Sanitized Approach for Learning Differentially Private Generators"

These supplementary materials include the privacy analysis (§A), the algorithm pseudocode (§B), the details of experiment setup (§C), and additional results (§D). Our source code is available on Github: `https://github.com/DingfanChen/GS-WGAN`.

## A   Privacy Analysis

The privacy cost ($\varepsilon$) computation including: *(i)* bounding the privacy loss for our gradient sanitization mechanism using RDP; *(ii)* applying analytical moments accountant of subsampled RDP [11] for a tighter upper bound on the RDP parameters; *(iii)* tracking the overall privacy cost: multiplying the RDP orders by the number of training iterations and converting the resulting RDP orders to an $(\varepsilon, \delta)$ pair (Definition 3.2 [7]). We below present the theoretical results.

**Theorem 4.1.** Each generator update step satisfies $(\lambda, 2B\lambda/\sigma^2)$-RDP where $B$ is the batch size.

*Proof.* Let $f = \text{clip}(\boldsymbol{g}_G^{\text{upstream}}, C)$, i.e., the clipped gradient before being sanitized. The sensitivity can be derived via the triangle inequality:

$$\Delta_2 f = \max_{S,S'} \|f(S) - f(S')\|_2 \leq 2C \tag{1}$$

with $C = 1$ in our case. Hence, we have $\mathcal{M}_{\sigma,C}$ is $(\lambda, 2\lambda/\sigma^2)$-RDP.
Each generator update step (which operates on a batch of data) can be expressed as

$$\hat{\boldsymbol{g}}_G = \frac{1}{B} \sum_{i=1}^{B} \mathcal{M}_{\sigma,C}(\nabla_{G(\boldsymbol{z}_i)} \mathcal{L}_G(\boldsymbol{\theta}_G)) \cdot J_{\boldsymbol{\theta}_G} G(\boldsymbol{z}_i; \boldsymbol{\theta}_G) \tag{2}$$

This can be seen as a composition of $B$ Gaussian mechanisms. Concretely, we want to bound the Rényi divergence $D_\lambda(\hat{\boldsymbol{g}}_G(S) \| \hat{\boldsymbol{g}}_G(S'))$ with $S, S'$ denoting the neighbouring datasets. We use the following properties of Rényi divergence [10]:
*(i)* Data-processing inequality : $D_\lambda(P_Y \| Q_Y) \leq D_\lambda(P_X \| Q_X)$ if the transition probabilities $A(Y|X)$ in the Markov chain $X \to Y$ is fixed.
*(ii)* Additivity : For arbitrary distributions $P_1, .., P_N$ and $Q_1, ..., Q_N$ let $P^N = P_1 \times \cdots \times P_N$ and $Q^N = Q_1 \times \cdots \times Q_N$. Then $D_\lambda(P^N \| Q^N) = \sum_{n=1}^{N} D_\lambda(P_n \| Q_n)$

Let $u$ and $v$ denote the output distribution of the sanitization mechanism $\mathcal{M}_{\sigma,C}$ when applied on $S$ and $S'$ respectively, and $h$ the post-processing function (i.e., multiplication with the local Jacobian). We have,

$$D_\lambda(\hat{\boldsymbol{g}}_G(S), \hat{\boldsymbol{g}}_G(S')) \leq D_\lambda\Big(h_1(u_1) * \cdots * h_B(u_B) \| h_1(v_1) * \cdots * h_B(v_B)\Big) \tag{3}$$

$$\leq D_\lambda\Big(\big(h_1(u_1), \cdots, h_B(u_B)\big) \| \big(h_1(v_1), \cdots, h_B(v_B)\big)\Big) \tag{4}$$

$$= \sum_b D_\lambda((h_b(u_b) \| h_b(v_b)) \tag{5}$$

$$\leq \sum_b D_\lambda(u_b \| v_b) \tag{6}$$

$$\leq B \cdot \max_b D_\lambda(u_b \| v_b) \tag{7}$$

$$\leq B \cdot 2\lambda/\sigma^2 \tag{8}$$

where (3)(4)(6) are based on the data-processing theorem; (5) follows from the additivity; and the last equation follows from the $(\lambda, 2\lambda/\sigma^2)$-RDP of $\mathcal{M}_{\sigma,C}$. $\qquad\square$

**Theorem A.1.** (RDP for Subsampled Mechanisms [11]) Given a dataset containing $n$ datapoints with domain $\mathcal{X}$ and a randomized mechanism $\mathcal{M}$ that takes an input from $\mathcal{X}^m$ for $m \leq n$, let the randomized algorithm $\mathcal{M} \circ \textbf{subsample}$ be defined as: *(i)* **subsample**: subsample without replacement $m$ datapoints of the dataset (with subsampling rate $\gamma = m/n$); *(ii)* apply $\mathcal{M}$: a randomized algorithm taking the subsampled dataset as the input. For all integers $\lambda \geq 2$, if $\mathcal{M}$ is $(\lambda, \epsilon(\lambda))$-RDP, then $\mathcal{M} \circ \textbf{subsample}$ is $(\lambda, \epsilon'(\lambda))$-RDP where

$$
\begin{aligned}
\epsilon'(\lambda) \leq &\frac{1}{\lambda-1} \log \left(1 + \gamma^2 \binom{\lambda}{2} \min \left\{4(e^{\epsilon(2)} - 1), e^{\epsilon(2)} \min \{2, (e^{\epsilon(\infty)} - 1)^2\}\right\}\right. \\
&\left. + \sum_{j=3}^{\lambda} \gamma^j \binom{\lambda}{j} e^{(j-1)\epsilon(j)} \min\{2, (e^{\epsilon(\infty)} - 1)^j\}\right)
\end{aligned}
$$

In practice, we adopt the official implementation of [11] [1] for computing the accumulated privacy cost (i.e., tracking the RDP orders and converting RDP to $(\varepsilon, \delta)$-DP).

# B  Algorithm

We present the pseudocode of our proposed method in Algorithm 1 (Centralized setup) and Algorithm 2 (Federated setup).

---

**Algorithm 1:** Centralized GS-WGAN Training

**Input:** Dataset $S$, subsampling rate $\gamma$, noise scale $\sigma$, warm-start iterations $T_w$, training iterations $T$, learning rates $\eta_D$ and $\eta_G$, the number of discriminator iterations per generator iteration $n_{dis}$, batch size $B$

**Output:** Differentially Private generator $G$ with parameters $\boldsymbol{\theta}_G$, total privacy cost $\varepsilon$

1 Subsample (without replacement) the dataset $S$ into subsets $\{S_k\}_{k=1}^{K}$ with rate $\gamma$ ($K = 1/\gamma$);
2 **for** $k$ **in** $\{1, ..., K\}$ **in parallel do**
3      Initialize non-private generator $\boldsymbol{\theta}_G^k$, discriminator $\boldsymbol{\theta}_D^k$ **for** *step* **in** $\{1, ..., T_w\}$ **do**
4          **for** $t$ **in** $\{1, ..., n_{dis}\}$ **do**
5              Sample batch $\{\boldsymbol{x}_i\}_{i=1}^{B} \subseteq S_k$ ;
6              Sample batch $\{\boldsymbol{z}_i\}_{i=1}^{B}$ with $\boldsymbol{z}_i \sim P_z$ ;
7              $\boldsymbol{\theta}_D^k \leftarrow \boldsymbol{\theta}_D^k - \eta_D \cdot \frac{1}{B} \sum_i \nabla_{\boldsymbol{\theta}_D^k} \mathcal{L}_D(\boldsymbol{\theta}_D^k; \boldsymbol{x}_i, G(\boldsymbol{z}_i; \boldsymbol{\theta}_G^k))$ ;
8          **end**
9          $\boldsymbol{\theta}_G^k \leftarrow \boldsymbol{\theta}_G^k - \eta_G \cdot \frac{1}{B} \sum_i \nabla_{\boldsymbol{\theta}_G^k} \mathcal{L}_G(\boldsymbol{\theta}_G^k; G(\boldsymbol{z}_i; \boldsymbol{\theta}_G^k), \boldsymbol{\theta}_D^k)$ ;
10      **end**
11      Initialize private generator $\boldsymbol{\theta}_G$ ;
12      **for** *step* **in** $\{1, ..., T\}$ **do**
13          Sample subset index $k \sim \mathcal{U}[1, K]$ ;
14          **for** $t$ **in** $\{1, ..., n_{dis}\}$ **do**
15              Sample batch $\{\boldsymbol{x}_i\}_{i=1}^{B} \subseteq S_k$ ;
16              Sample batch $\{\boldsymbol{z}_i\}_{i=1}^{B}$ with $\boldsymbol{z}_i \sim P_z$ ;
17              $\boldsymbol{\theta}_D^k \leftarrow \boldsymbol{\theta}_D^k - \eta_D \cdot \frac{1}{B} \sum_i \nabla_{\boldsymbol{\theta}_D^k} \mathcal{L}_D(\boldsymbol{\theta}_D^k; \boldsymbol{x}_i, G(\boldsymbol{z}_i; \boldsymbol{\theta}_G))$ ;
18          **end**
19          $\boldsymbol{\theta}_G \leftarrow \boldsymbol{\theta}_G - \eta_G \cdot \frac{1}{B} \sum_i \mathcal{M}_{\sigma,C}(\boldsymbol{\theta}_G; G(\boldsymbol{z}_i; \boldsymbol{\theta}_G), \boldsymbol{\theta}_D^k) \cdot \boldsymbol{J}_{\boldsymbol{\theta}_G} G(\boldsymbol{z}_i; \boldsymbol{\theta}_G)$ ;
20          Accumulate privacy cost $\varepsilon$ ;
21      **end**
22 **end**
23 **return** Generator $G(\cdot; \boldsymbol{\theta}_G)$, privacy cost $\varepsilon$

**Algorithm 2:** Federated (Decentralized) GS-WGAN Training

**Input:** Client index set $\{1, ..., K\}$, noise scale $\sigma$, warm-start iterations $T_w$, training iterations $T$, learning rates $\eta_D$ and $\eta_G$, the number of discriminator iterations per generator iteration $n_{dis}$, batch size $B$

**Output:** Differentially Private generator $G$ with parameters $\boldsymbol{\theta}_G$, total privacy cost $\varepsilon$

1 **for** each client $k$ **in** $\{1, ..., K\}$ **in parallel do**
2     `ClientWarmStart`$(k)$
3 **end**
4 Initialize private generator $\boldsymbol{\theta}_G$ ;
5 **for** $step$ **in** $\{1, ..., T\}$ **do**
6     Sample subset index $k \sim \mathcal{U}[1, K]$ ;
7     **for** $t$ **in** $\{1, ..., n_{dis}\}$ **do**
8        Sample batch $\{z_i\}_{i=1}^B$ with $z_i \sim P_z$ ;
9        $\{\hat{\boldsymbol{g}}_i^{\text{up}}\}_{i=1}^B \leftarrow$ `ClientUpdate`$(k, G(z_i; \boldsymbol{\theta}_G))$
10     **end**
11     $\boldsymbol{\theta}_G \leftarrow \boldsymbol{\theta}_G - \eta_G \cdot \frac{1}{B} \sum_i \hat{\boldsymbol{g}}_i^{\text{up}} \cdot \boldsymbol{J}_{\boldsymbol{\theta}_G} G(z_i; \boldsymbol{\theta}_G)$ ;
12     Accumulate privacy cost $\varepsilon$ ;
13 **end**
14 **return** Generator $G(\cdot ; \boldsymbol{\theta}_G)$, privacy cost $\varepsilon$

---

16 **Procedure** `ClientWarmStart`$(k)$
17     Get local dataset $S_k$ ;
18     Initialize local generator $\boldsymbol{\theta}_G^k$, discriminator $\boldsymbol{\theta}_D^k$ ;
19     **for** $step$ **in** $\{1, ..., T_w\}$ **do**
20        **for** $t$ **in** $\{1, ..., n_{dis}\}$ **do**
21           Sample batch $\{x_i\}_{i=1}^B \subseteq S_k$ ;
22           Sample batch $\{z_i\}_{i=1}^B$ with $z_i \sim P_z$ ;
23           $\boldsymbol{\theta}_D^k \leftarrow \boldsymbol{\theta}_D^k - \eta_D \cdot \frac{1}{B} \sum_i \nabla_{\boldsymbol{\theta}_D^k} \mathcal{L}_D(\boldsymbol{\theta}_D^k; x_i, G(z_i; \boldsymbol{\theta}_G^k))$ ;
24        **end**
25        $\boldsymbol{\theta}_G^k \leftarrow \boldsymbol{\theta}_G^k - \eta_G \cdot \frac{1}{B} \sum_i \nabla_{\boldsymbol{\theta}_G^k} \mathcal{L}_G(\boldsymbol{\theta}_G^k; G(z_i; \boldsymbol{\theta}_G^k), \boldsymbol{\theta}_D^k)$ ;
26     **end**

---

28 **Procedure** `ClientUpdate`$(k, G(z_i; \boldsymbol{\theta}_G))$
29     Get local dataset $S_k$, local discriminator $D(\cdot ; \boldsymbol{\theta}_D^k)$ ;
30     Sample batch $\{x_i\}_{i=1}^B \subseteq S_k$ ;
31     $\boldsymbol{\theta}_D^k \leftarrow \boldsymbol{\theta}_D^k - \eta_D \cdot \frac{1}{B} \sum_i \nabla_{\boldsymbol{\theta}_D^k} \mathcal{L}_D(\boldsymbol{\theta}_D^k; x_i, G(z_i; \boldsymbol{\theta}_G))$ ;
32     **return** $\mathcal{M}_{\sigma,C}(\boldsymbol{\theta}_G; G(z_i; \boldsymbol{\theta}_G), \boldsymbol{\theta}_D^k)$

## C  Experiment Setup

### C.1  Hyperparameters

We adopt the hyperparameters setting in [3] for the GAN training, and list below the hyperparameters relevant for privacy computation.

**Centralized Setting.** We use by default a subsampling rate of $\gamma$=1/1000, noise scale $\sigma$=1.07, pretraining (warm-start) for 2K iterations and subsequently training for 20K iterations.

**Federated Setting.** We use by default a noise scale $\sigma$=1.07, pretraining (warm-start) for 2K iterations and subsequently training for 30K iterations.

### C.2  Datasets

**Centralized Setting.** **MNIST** [5] and **Fashion-MNIST** [12] datasets contain 60K training images and 10K testing images. Each image has dimension $28 \times 28$ and belongs to one of the 10 classes.

**Federated Setting.** **Federated EMNIST** [2] dataset contains $28 \times 28$ gray-scale images of hand-written letters and numbers, grouped by user. The entire dataset contains 3400 users with 671,585 training examples and 77,483 testing examples. Following [1], the users are filtered by the prediction accuracy of a 36-class (10 numeric digits + 26 letters) CNN classifier. For evaluating the sample quality, we train GAN models on the users' data which yields classification accuracy $\geq 93.9\%$ (866 users); For simulating the debugging task, we randomly choose $50\%$ of the users and pre-process their data by flipping the pixel intensities. To mimic the real-world situation where the server is blind to the erroneous pre-processing, users with low classification accuracy $\leq 88.2\%$ are selected (2136 users) as they are suspected to be affected by erroneous flipping (with bug). Note that only a fraction of them is indeed affected by the bug (1720 with bug, 416 without bug). This has the realistic property that the client data is non-IID and poses additional difficulties in the GAN training.

## C.3 Evaluation Metrics

In line with previous literature, we use Inception Score (IS) [6, 9] and Frechet Inception Distance (FID) [4] for measuring sample quality, and classification accuracy for evaluating the usefulness of generated samples. We present below a detailed explanation of the evaluation metrics we adopted in the experiments.

**Inception Score (IS).** Formally, the IS is defined as follows,

$$\text{IS} = \exp\left(\mathbb{E}_{\boldsymbol{x} \sim G(\boldsymbol{z})} D_{KL}(P(y|\boldsymbol{x}) \| P(y))\right)$$

which corresponds to exponential of the KL divergence between the conditional class $P(y|\boldsymbol{x})$ and the marginal class distribution $P(y)$, where both $P(y|\boldsymbol{x})$ and $P(y)$ are measured by the output distribution of a pre-trained classifier when passing the generated samples as input. Intuitively, the IS should exhibit a high value if $P(y|\boldsymbol{x})$ has low entropy (i.e., the generated images are sharp and contain clear objects) and $P(y)$ is of high entropy (i.e., the generated samples have a high diversity covering all the different classes). In our experiments, we use pre-trained classifiers on the real datasets (with test accuracy equals to 99.25%, 93.75%, 92.16% on the MNIST, Fashion-MNIST and Federated EMNIST dataset respectively) [2] for computing the IS.

**Frechet Inception Distance (FID).** The FID is formularized as follows,

$$\text{FID} = \|\mu_r - \mu_g\|^2 + \text{tr}(\Sigma_r + \Sigma_g - 2(\Sigma_r \Sigma_g)^{1/2})$$

where $\boldsymbol{x}_r \sim \mathcal{N}(\mu_r, \Sigma_r)$ and $\boldsymbol{x}_g \sim \mathcal{N}(\mu_g, \Sigma_g)$ are the 2048-dimensional activations of the Inception-v3 pool3 layer for real and generated samples respectively. A lower FID value indicates a smaller discrepancy between the real and generated samples, which corresponds to a better sample quality and diversity. Following previous works [3] , we rescale the images and convert them to RGB by repeating the grayscale channel three times before inputting them to the Inception network.

**Classification Accuracy.** We consider the following classification models in our experiments: Multi-layer Perceptron (MLP), Convolutional Neural Network (CNN), AdaBoost (adaboost), Bagging (bagging), Bernoulli Naive Bayes (bernoulli nb), Decision tree (decision tree), Gaussian Naive Bayes (gaussian nb), Gradient Boosting (gbm), Linear Discriminant Analysis (lda), Linear Support Vector Machine (linear svc), Logistic Regression (logistic reg), Random Forest (random forest), and XGBoost (xgboost). For implementing the CNN model, we use two hidden layers (with dropout) each containing 32 and 64 kernels and apply ReLU as the activation function. For implementing the MLP, we use one hidden layer with 100 neurons and set ReLU as the activation function. All the other classification models are implemented using the default hyperparameters supplied by the scikit-learn [8] package.

## C.4 Baseline Methods

We present more details about the implementation of the baseline methods. In particular, we provide the default value of the privacy hyperparameters below.

**DP-Merf (AE)** [4] We use as default a batch size=500 ($\gamma$=1/120), noise scale $\sigma$=0.588, training iteration=600 (epoch=5) for implementing DP-Merf, and batch size=500, noise scale $\sigma$=0.686, training iteration=2040 (epoch=17) for implementing DP-Merf AE.

**DP-SGD GAN** [5] We set the default hyper-parameters as follows: gradient clipping bound $C$=1.1, noise scale $\sigma$=2.1, batch size=600, training iterations=30K.

**G-PATE** We use 2000 teacher discriminators with batch size of 30 and set noise scales $\sigma_1$=600 and $\sigma_2$=100, consensus threshold $T$=0.5. A random projection with projection dimension=10 is applied.

**PATE-GAN** [6] When extending PATE-GAN to high-dimensional image datasets, we observe that after a few iterations, the generated samples are classified as fake by all teacher discriminators and the learning signals (gradients) for student discriminator and the generator vanish. Consequently, the training stuck at the early stage where the losses remain unchanged and no progress can be observed. While this issue is well resolved by careful design of the prior distribution, as reported in the original paper, we find that this technique has a limited effect when applied to the high-dimensional image dataset. In addition, we make the following attempts to address this issue: *(i)* changing the network initialization *(ii)* increasing (or decreasing) the network capacity of the student discriminator, the teacher discriminators, and the generator *(iii)* increasing the number of iterations for updating the student discriminator and/or the generator. Despite some progress in preserving the gradients for larger iterations, none of the above attempts successfully eliminate the issue, as the training inevitably gets stuck within 1K iterations.

# D   Additional Results

**Effects of gradient clipping.** We show in Figure A1 the gradient norm distribution before and after gradient clipping. The clipping bound is set to be 1.1 for DP-SGD and 1 for our method. In contrast to DP-SGD, the clipping operation distorts less information in our framework, witnessed by a much smaller difference in the average gradient norm before and after the clipping. Moreover, the gradients used in our method exhibit much less variance both before and after the clipping compared with DP-SGD.

| (a) DP-SGD (before) | (b) DP-SGD (after) | (c) Ours (before) | (d) Ours (after) |

**Figure A1:** Effects of gradient clipping.

**Comparison to Baselines.** We provide the detailed quantitative results in Table A1 and A2, which are supplementary to Table 1 in the main paper. We show in parentheses the calibrated accuracy, i.e., the absolute accuracy of each classifier trained on generated data divided by the accuracy when trained on real data. The results are averaged over five runs.

**Privacy-utility Curves.** We show in Figure A2 the privacy-utility curves of different methods when applied to the Fashion-MNIST dataset. We evaluate over three runs and show the corresponding mean and standard deviation. Similar to the results shown in Figure 4 in the main paper, our method achieves a consistent improvement over prior methods across a broad range of privacy budget $\varepsilon$.

      eatures-for-Synthetic-Data-Generation
[5]   https://github.com/reihaneh-torkzadehmahani/DP-CGAN
[6]   https://bitbucket.org/mvdschaar/mlforhealthlabpub/src/2534877d99c8fdf19cbade1605
      7990171e249ef3/alg/pategan/

|  | Real | GAN (non-private) | G-PATE | DP-SGD GAN | DP-Merf | DP-Merf AE | Ours |
|---|---|---|---|---|---|---|---|
| MLP | 0.98 | 0.84 (85%) | 0.25 (26%) | 0.60 (61%) | 0.63 (64%) | 0.54 (55%) | 0.79 (81%) |
| CNN | 0.99 | 0.84 (85%) | 0.51 (52%) | 0.64 (65%) | 0.63 (64%) | 0.68 (69%) | 0.80 (81%) |
| adaboost | 0.73 | 0.28 (39%) | 0.11 (16%) | 0.32 (44%) | 0.38 (52%) | 0.21 (29%) | 0.21 (29%) |
| bagging | 0.93 | 0.46 (49%) | 0.36 (38%) | 0.44 (47%) | 0.43 (46%) | 0.33 (35%) | 0.45 (48%) |
| bernoulli nb | 0.84 | 0.80 (95%) | 0.71 (84%) | 0.62 (74%) | 0.76 (90%) | 0.50 (60%) | 0.77 (92%) |
| decision tree | 0.88 | 0.40 (45%) | 0.13 (14%) | 0.36 (41%) | 0.29 (33%) | 0.27 (31%) | 0.35 (40%) |
| gaussian nb | 0.56 | 0.71 (126%) | 0.61 (110%) | 0.37 (66%) | 0.57 (102%) | 0.17 (30%) | 0.64 (114%) |
| gbm | 0.91 | 0.50 (55%) | 0.11 (12%) | 0.45 (49%) | 0.36 (40%) | 0.20 (22%) | 0.39 (43%) |
| lda | 0.88 | 0.84 (95%) | 0.60 (68%) | 0.59 (67%) | 0.72 (82%) | 0.55 (63%) | 0.78 (89%) |
| linear svc | 0.92 | 0.81 (88%) | 0.24 (26%) | 0.56 (61%) | 0.58 (63%) | 0.43 (47%) | 0.76 (83%) |
| logistic reg | 0.93 | 0.83 (90%) | 0.26 (28%) | 0.60 (65%) | 0.66 (71%) | 0.55 (59%) | 0.79 (85%) |
| random forest | 0.97 | 0.39 (41%) | 0.33 (34%) | 0.63 (65%) | 0.66 (68%) | 0.45 (46%) | 0.52 (54%) |
| xgboost | 0.91 | 0.44 (49%) | 0.15 (16%) | 0.60 (66%) | 0.70 (77%) | 0.54 (59%) | 0.50 (55%) |
| Average | 0.88 | 0.63 (71%) | 0.34 (40%) | 0.52 (59%) | 0.57 (66%) | 0.42 (47%) | 0.60 (69%) |

**Table A1:** Classification accuracy on MNIST ($\varepsilon = 10, \delta = 10^{-5}$).

|  | Real | GAN (non-private) | G-PATE | DP-SGD GAN | DP-Merf | DP-Merf AE | Ours |
|---|---|---|---|---|---|---|---|
| MLP | 0.88 | 0.77 (88%) | 0.30 (34%) | 0.50 (57%) | 0.56 (64%) | 0.56 (64%) | 0.65 (74%) |
| CNN | 0.91 | 0.73 (80%) | 0.50 (54%) | 0.46 (51%) | 0.54 (59%) | 0.62 (68%) | 0.64 (70%) |
| adaboost | 0.56 | 0.41 (74%) | 0.42 (75%) | 0.21 (38%) | 0.33 (59%) | 0.26 (46%) | 0.25 (45%) |
| bagging | 0.84 | 0.57 (68%) | 0.38 (45%) | 0.32 (38%) | 0.40 (47%) | 0.45 (54%) | 0.47 (56%) |
| bernoulli nb | 0.65 | 0.59 (91%) | 0.57 (88%) | 0.50 (77%) | 0.62 (95%) | 0.54 (83%) | 0.55 (85%) |
| decision tree | 0.79 | 0.53 (67%) | 0.24 (30%) | 0.33 (42%) | 0.25 (32%) | 0.36 (46%) | 0.40 (51%) |
| gaussian nb | 0.59 | 0.55 (93%) | 0.57 (97%) | 0.28 (47%) | 0.59 (100%) | 0.12 (20%) | 0.48 (81%) |
| gbm | 0.83 | 0.44 (53%) | 0.25 (30%) | 0.38 (46%) | 0.27 (33%) | 0.30 (36%) | 0.38 (46%) |
| lda | 0.80 | 0.77 (96%) | 0.55 (69%) | 0.55 (69%) | 0.67 (84%) | 0.65 (81%) | 0.67 (84%) |
| linear svc | 0.84 | 0.77 (91%) | 0.30 (36%) | 0.39 (46%) | 0.46 (55%) | 0.40 (48%) | 0.65 (77%) |
| logistic reg | 0.84 | 0.76 (90%) | 0.35 (42%) | 0.51 (61%) | 0.59 (70%) | 0.50 (60%) | 0.68 (81%) |
| random forest | 0.88 | 0.69 (78%) | 0.33 (37%) | 0.51 (58%) | 0.61 (69%) | 0.55 (63%) | 0.54 (61%) |
| xgboost | 0.83 | 0.65 (78%) | 0.49 (59%) | 0.52 (63%) | 0.62 (75%) | 0.55 (66%) | 0.47 (57%) |
| Average | 0.79 | 0.61 (77%) | 0.40 (54%) | 0.42 (53%) | 0.50 (65%) | 0.45 (56%) | 0.53 (67%) |

**Table A2:** Classification accuracy on Fashion-MNIST ($\varepsilon = 10, \delta = 10^{-5}$).

**Figure A2:** Privacy-utility trade-off on Fashion-MNIST with $\delta = 10^{-5}$. (Left: Effects of noise scale. Right: Effects of Iterations.)

## Footnotes

[1] `https://github.com/yuxiangw/autodp`

[2]    https://github.com/ChunyuanLI/MNIST_Inception_Score

[3]    https://github.com/google/compare_gan

[4]   https://github.com/frhrdr/Differentially-Private-Mean-Embeddings-with-Random-F