[Reviews · NeurIPS 2020]

Review 1

Summary and Contributions: The main contribution of the paper is to observe that in an application of DP GAN for private synthetic data generation, only generator needs to be private, so algorithmically we can move the DP-SGD step (i.e. clip gradients and add noise) from being applied at the discriminator but not generator (as in standard DP-GAN) to applied at the generator but not discriminator. This simple change allows for more complicated ideas to be applied on the discriminator side. Particularly, it allows for a more complex architecture and warm-starting the discriminator (line 212-221). It also applies more seamlessly with the federated setting, where there are multiple discriminators and the gradients are aggregated before being fed into the generator. Experiments show empirical results that beat the existing works. Edits to author's feedback: - Thank you for clarifying the theoretical underpinning of WGAN and the difference between your approach to the existing WGAN. - I would still suggest adding a comparison between your work with standard WGAN (and properly cite them) in related work and a line or two explaining specifically how yours differ from WGAN (i.e. that you can bound gradient norm but previous one can't) - I still suggest accept due to the connection of Wasserstein-1 metric loss to bounding the gradient norm in a new simpler way to sanitize gradient.

Strengths: I like the simplicity of the idea. It is an observation to move clipping and adding noise from discriminator to generator, but it gives many applications. With simplicity, it comes with good and extensive empirical results. It improves baseline significantly in all metrics. It now makes a good MNIST DP synthetic data possible with eps=10 (Figure 3). Due to its simple idea and simple change to DP-GAN, and higher flexibility to design and train discriminator (or discriminators in federated case), I think practitioners should find this idea useful.

Weaknesses: The only case that the main idea causes limitation would be if discriminator is needed for some public use, which I think would be rare, so I don't see a significant limitation with this approach. There are some other weaknesses/concerns, but they are listed in corresponding sections below.

Correctness: Theoretically there aren't any new contributions. The proof of Theorem 4.1, to me, is a simple restating of standard use of RDP in DP-SGD (the only change is to say that J_local is non-private information, instead of using post-processing to claim privacy for generator's gradient as in standard DP-GAN), and is (somewhat trivially) correct. Theorem A.1 is also just a restatement. Empirical methods seem correct to the best of my knowledge.

Clarity: I can follow the main points and the flow of the paper relatively easy. There are some typos here and there, and can be fixed by some more proofreading, but are not significant to be distracting. Line 142-170. I am not sure what "bounding" the norm means in this section: it seems empirically rather than any theoretical claim, right? There is some reason that Wasserstein objective bound the gradient norm better (line 151-155), i.e. the norm of the last layer's gradient with respect to the loss, but I guess no reason to believe the norm in later layer will be bounded by 1 (let alone multiple layers)? The only evidence I see is empirical, or is there other argument I am missing? But this raises the question: is C=1 (line 167) something we should be that confident about due to some theoretical reasoning (as you claim to eliminate hyperparameter search), or would it change again when new architecture, dataset, and application come? I think other works with WGAN objectives (see below) have used other C's, so it seems there's still a need to tune C? Note: Theorem 4.1 on privacy analysis has nothing to do with using Wasserstein objective; any objective works as long as we clip and add noise the gradient as usual in DP-SGD to the generator by eq(11). So I am also not sure of the connection from Wasserstein section to the theorem (line 168-170). Related to this, line 294 uses the word "estimate" which is has a technical/statistical meaning, but I don't see any kind of estimate resulting from using Waserstein objective in the paper, which can be misleading.

Relation to Prior Work: Is the proposed eq(11) actually an identical optimization-step rule for the generator as standard DP-GAN? In standard DP-GAN, we clip and add noise to the discriminator's gradient (G^upstream), then back propagate without noise (multiply by J_local). If so, would it be that the empirical improvement comes primarily in better and more complex discriminator (since generator architecture probably suffer the same noise level as in previous DP-GAN work anyway)? It is unclear whether you claim both gradient-sanitization and Wasserstein GAN or just gradient-sanitization to be your contribution from line 46. A section on Wasserstein objective (line 142-170) without references to DP-WGAN in this section suggests the former. However, there are previous DP-WGAN such as https://arxiv.org/pdf/1802.06739.pdf, https://arxiv.org/pdf/1801.01594.pdf, https://arxiv.org/abs/1901.02477, and https://arxiv.org/abs/1912.03250 (the first two you also cited). The first one also has a theoretical guarantee (though pretty loose) of DP WGAN through weight clipping, but the other three only uses Wasserstein objectives due to its empirical performance. If there is no additional contribution besides empirical observations, the new contribution would be a clearer plot of gradient norms of DP-SGD vs WGAN, possibly useful and good to know, but can be marginal.

Reproducibility: No

Additional Feedback: line 34: note that DP can work for a large network for large data (e.g. https://arxiv.org/abs/1710.06963 applies DP to 1M parameters recurrent network). Figure 4(c): is there a reason why epsilon<2 is not in the Figure? How do methods do with high-privacy regime? Line 312-313: there is no code provided, but the authors claim in reproducibility checklist part 4 to provide code, README, etc. I also don't see any details of variation/error bars nor computing infrastructure details which authors claim to provide in reproducibility in the paper. Minor comments: eq(10): notation J used. It may be good to say it is Jacobian when first used it (not as common as gradient notation?). Figure 2: I would appreciate experimental details/setup of Figure 2. Is it the same as in the paper? I would also like an explanation on how to read 2(a), 2(c) and similar plots. Line 196: dataset --> datasets (last word of the sentence) Table 1: maybe worth mentioning what arrows up/down mean L. 238: space before the parenthesis "(Table 1" is missing L. 274: space after [3]. is missing Appendix: - line 33: I believe [11] is the incorrect citation


Review 2

Summary and Contributions: This paper proposes a differentially private algorithm for generating synthetic datasets. Based on WGAN-GP, the authors propose to add noise to the generator gradient and only release the generator to the public. The reason for choosing WGAN-GP is because the gradient penalty makes the generator to have intrinsic low sensitivity. The authors demonstrate the sample quality performance on MNIST and compare with the state of the art dp generative models.

Strengths: This paper provides a practical algorithm with privacy guarantees. Compared to existing methods, this paper achieves better privacy utility trade-off. The experiments are convincing.

Weaknesses: The improvement relies on the WGAN-GP. I am not sure if the performance would still be good if we use other GAN architectures.

Correctness: I think privacy guarantees is correct. The experiment might not be a fair comparison since this paper is using a powerful generator from BigGAN.

Clarity: It is overall well written. I am confused about Figure 4 (b)and Figure A2. I think epsilon could be considered as privacy budget. Calling it noise scale is incorrect since the actual noise scale is also related to the sensitivity. Number of iterations and other hyperparameters should be chosen according to the same privacy budget.

Relation to Prior Work: Yes. This paper has compared with a few DP based generative model baselines

Reproducibility: Yes

Additional Feedback: Post rebuttal: Author feedback has answered my questions. I will retain my score. I agree that this paper provides a practical and simple algorithm though the idea is not groundbreaking.


Review 3

Summary and Contributions: This paper develops a novel GAN that enables to release a sanitized version of the sensitive data with privacy guarantees. That method uses Wasserstein objectives with gradient penalty term to improve the gradient compared to the other approaches during training. Another advantage of the proposed method is to get rid of the searching of DP hyperparameters. And that approach natuarally extends to both centralized and decentralized settings.

Strengths: The method presented in this paper has several advantages, one is to provide a more accurate gradient information for the same level of privacy protection with the state of the art. Besides, it doesn’t require to search for gradient clipping values. Those contributions lead to the better performance of that approach and it is the most significant strength of the paper. Besides, the paper has an extensive set of experiments that answers every questions that has brought in the methods section and they are well organized.

Weaknesses: It would be nice to add the differences of the proposed approach with the state of the art algorithms that are given in the related work section explicitly and clearly. In the current version, it is clear the contribution of the proposed approach, but it is not clear if another existing approach contributed in the same way before. Apart form that, the background section only contains the very basic Definitions and Theorems of differential privacy literature. In my opinion, it would be better to see at least some connections between those and the proposed method in the main paper instead of Section A - Privacy Analysis in the supplementary material. That could provide some intuition to the readers. == Update after rebuttal == I appreciate for the response. They clarified my concerns, thank you.

Correctness: I checked both the equations in the main paper and the analysis in the supplementary. To the best of my knowledge, the methods and the empirical methodology are correct.

Clarity: The paper is mostly well written. But the structure of the paper at some sections could be modified. I added my comments in the weaknesses part which is basically for Related Work and the Background sections.

Relation to Prior Work: That was the main weakness of the paper in my opinion. I would prefer to see the differences more explicitly and in the bigger picture. Currently, the paper provides the differences more in the technical details.

Reproducibility: Yes

Additional Feedback:


Review 4

Summary and Contributions: This paper extends WGAN to reserve the data privacy, namely the generator will sanitize the data with DP guarantees but still allow to train a reasonable model on the sanitized data. This paper proposes to use the Wasserstein-1 metric as the loss, so the gradient norms are around 1, therefore the following gradient clip step, which is for DP guarantees, will not affect the gradients too much. Empirical results are reported on MNIST and FashionMNIST.

Strengths: The idea of using a proper loss to reduce the damage from gradient clipping while still preserve DP sounds interesting. It not only improves the accuracy but also simplifies the usage as we don't need to specify the clipping threshold anymore.

Weaknesses: Only two simple datasets (MNIST and FashionMNIST) are evaluated. I'm wondering if the proposed method still works on more complicated datasets, such as on nature images. As the loss function is changed, it's not very clear if it will affect the training time or complicate the training a lot.

Correctness: I scanned the Appendix A (proof for theorem 4.1), it seems correct to me.

Clarity: Yes.

Relation to Prior Work: Yes.

Reproducibility: Yes

Additional Feedback:

[Author Response · NeurIPS 2020]

We thank the reviewers for their constructive feedback! We have detailed our response below and will improve the
manuscript accordingly. We are pleased that reviewers appreciate our approach 'interesting' (**R#5**) , 'not only improves
the accuracy but also simplifies the usage '(**R#5**), 'improves baseline significantly in all metrics' (**R#2**) and 'gives many
applications' **R#2**), and that the experiments are 'convincing' (**R#3**), 'extensive' (**R#2**,**R#4**), 'well organized' (**R#4**)
and 'answers every question brought in the methods section' (**R#4**).

[**R#2**] [**"bounding" the norm empirically rather than theoretical? Line 294 uses the word "estimate" which has
a technical/statistical meaning**] It is theoretical: norm of $g_G^{\text{upstream}}$ is 1 almost everywhere (Line 151-155). We will
further clarify. The proof can be found in [17] (Proposition 1 and Corollary 1, where $g_G^{\text{upstream}}$ corresponds to $\nabla f^*$).

[**Would C=1 change for multiple layers? new datasets, architectures, etc.?**] No (Line 136-139). Note that we only
need to bound $\|g_G^{\text{upstream}}\|_2$. $C = 1$ follows from the theoretical property of Wasserstein objective and is independent on
all other factors including architectures and datasets.

[**Other works using WGAN used other C values ... need to tune C?**] Our approach does not require tuning $C$.
In previous DP-SGD GAN framework, the discriminator parameter gradients $\nabla_{\boldsymbol{\theta}_D}\mathcal{L}_D(\boldsymbol{\theta}_D)$ are sanitized, instead of
$g_G^{\text{upstream}}$. In comparison, $\nabla_{\boldsymbol{\theta}_D}\mathcal{L}_D(\boldsymbol{\theta}_D)$ do not have any bounded norm guarantee (empirically the gradient norms exhibit
a heavy-tailed distribution with high variance as shown in Figure 2) and previous works need to tune $C$ carefully.

[**Not sure of the connection from Wasserstein section to the theorem**] The DP privacy analysis relies on a sensitivity
value that depends on $C$. We show that by selectively applying sanitization to a necessary and sufficient subset of
gradients for preserving privacy, we are able to exploit the theoretical property of WGAN and bound the sensitivity
value without extensive tuning of $C$, while introducing almost no clipping bias.

[**Is Eq (11) an identical optimization-step rule for the generator as standard DP-GAN?**] No. As mentioned above,
the standard DP-SGD GAN sanitize different gradients as done in our framework, and their DP sanitization process
can *not* be modelled by Eq (11) ( $g_G^{\text{upstream}}$ does depend on $\nabla_{\boldsymbol{\theta}_D}\mathcal{L}_D(\boldsymbol{\theta}_D)$ but the dependence is represented by the
discriminator network, which can not be explicitly formulized).

[**The empirical improvement comes primarily in better and more complex discriminator?**] We attribute the
improvements to a combination of complex discriminator and our approach which allows optimizing discriminator's
parameters *without* clipping its gradients and consequently leads to a better trained discriminator. In contrast, standard
DPGANs introduce much larger clipping bias than our method, which also explain our superior performance.

[**Whether both gradient-sanitization and Wasserstein GAN or just gradient-sanitization is the contribution**] In
comparison to prior works on DPGAN, our proposed gradient sanitization scheme enable us to exploit the theoretical
property of WGAN. To the best of our knowledge, we are the first that consider the intrinsic property of WGAN and
propose a framework that can use this property for obtaining a theoretically grounded choice of $C$.

[**Line 34: note that DP can work for a large network for large data**] We thank for the reference and will rephrase.

[**Figure 4(c): why epsilon<2 is not in the Figure?**] We will make it consistent. Note that numerical issues arise from
the required noise scale for small epsilon values.

[**Release of source code**] As stated in Line 312-313 and Appendix Line 1-3, we will submit our source code together
with our final version and also publish our code and setups at the time of publication.

[**R#3**] [**Experiments: what if using other GAN architectures**] We conduct additional experiments and show that
our method with a basic DCGAN structure still yields consistent improvement over prior works, as shown in Table 1
("Previous Best" denotes the best score achieved by previous methods; The new experiments are repeated twice and the
average is reported). Moreover, we would like to emphasize that the usage of large network is prohibited by previous
   methods due to heavy hyperparameter search and a large clipping bias, which are well addressed by our framework.

|  |  | IS↑ | FID ↓ | MLP Acc↑ | CNN Acc↑ | Avg Acc↑ | Calibrated Acc↑ |
|---|---|---|---|---|---|---|---|
| MNIST | Previous Best | 4.76 | 161.11 | 0.63 | 0.68 | 0.57 | 66% |
|  | Ours (DCGAN) | **8.74** | **75.83** | **0.79** | **0.79** | **0.59** | **68%** |
| Fashion-MNIST | Previous Best | 3.68 | 205.78 | 0.56 | 0.62 | **0.51** | **65%** |
|  | Ours (DCGAN) | **5.59** | **134.74** | **0.67** | **0.66** | **0.51** | **65%** |

Table 1: Quantitative Results on MNIST and Fashion-MNIST ($\varepsilon = 10, \delta = 10^{-5}$).

[**Meaning of figure titles** ] The title named 'noise scale' means that we control the $\varepsilon$ by only changing the noise scale
while keeping all other factors fixed to be their default values. As stated in Line 243-246, we indeed consider the
different factors that affect $\varepsilon$ and investigate all of them .

[**R#4**] [**Bigger picture of differences between proposed approach and existing works**] We will enrich our discussion
and include a high-level comparison between our proposed method and the existing works.

[**Background section contains only definitions ... add more connections and intuitions**] We will improve the
background section to better equip readers with intuitions required to understand our approach.

[**R#5**] [**Using more complicated datasets, such as on nature images**] It would be ideal to evaluate on complex
high-dimensional data. However, due to the data and model complexity, it is currently challenging to obtain reasonable
performances for low privacy costs ($\varepsilon \leq 10$). Consequently, we use the same datasets and evaluation methodology as
prior works, in which we show substantial improvements and achieve state-of-the-art performances.

[Meta-Review · NeurIPS 2020]

The paper proposes a simple yet elegant change to make DP GANs significantly more accurate and presents the finding clearly. All reviewers agree that the technical novelty is not especially big, but as the method appears practically significant, the paper should be accepted.